# Autoinflammatory patients with Golgi-trapped CDC42 exhibit intracellular trafficking defects leading to STING hyperactivation and ER stress

Alberto Iannuzzo [1], Selket Delafontaine [2,3], Rana El Masri[1,13], Rachida Tacine[1], Giusi Prencipe[4], Masahiko Nishitani-Isa [5], Rogier T. A. van Wijck[6], Farzana Bhuyan[7], Adriana A. de Jesus Rasheed [7], Simona Coppola [8], Paul L. A. van Daele[9,10], Antonella Insalaco[11], Raphaela Goldbach-Mansky [7], Takahiro Yasumi [5], Marco Tartaglia [12], Isabelle Meyts[2,3] & Jérôme Delon [1]✉

Most autoinflammatory diseases are caused by mutations in innate immunity genes. Previously, four variants in the RHO GTPase CDC42 were discovered in patients affected by syndromes generally characterized by neonatal-onset of cytopenia and auto-inflammation, including hemophagocytic lymphohistiocytosis and rash in the most severe form (NOCARH syndrome). However, the mechanisms responsible for these phenotypes remain largely elusive. Here, we show that the recurrent p.R186C CDC42 variant, which is trapped in the Golgi apparatus, elicits a block in both anterograde and retrograde transports. Consequently, it favours STING accumulation in the Golgi in a COPI-dependent manner. This is also observed for the other Golgi-trapped p.*192 C*24 CDC42 variant, but not for the p.Y64C and p.C188Y variants that do not accumulate in the Golgi. We demonstrate that the two Golgi-trapped CDC42 variants are the only ones that exhibit overactivation of the STING pathway and the type I interferon response, and elicit endoplasmic reticulum stress. Consistent with these results, patients carrying Golgi-trapped CDC42 mutants present very high levels of circulating IFNα at the onset of their disease. In conclusion, we report further mechanistic insights on the impact of the Golgi-trapped CDC42 variants. This increase in STING activation provides a rationale for combination treatments for these severe cases.

Autoinflammatory diseases (AIDs) are driven by a hyperactivation of the innate immune system. They include clinically distinct disorders that present with systemic sterile inflammation, urticaria-like rashes, and disease-specific organ inflammation and damage. Some AIDs present with adaptive immune dysregulation, immunodeficiencies or with features of hyperinflammation, the latter including cytopenias, hyperferritinemia, hepatosplenomegaly, and high IL-18 levels[1–3]. Most AIDs are caused by germline mutations in genes that encode key regulators of innate immunity.

Our laboratory and others have previously reported heterozygous mutations in *CDC42*, encoding for a RHO GTPase which controls various cellular functions, including proliferation, migration, polarization, adhesion, cytoskeletal modifications and transcriptional regulation[4]. Three types of CDC42 variants localized in the plasma membrane

anchoring C-terminal region of CDC42 (c.556 C > T, p.R186C; c.563 G > A, p.C188Y and c.576 A > C, p.*192 C*24) have been found in patients with neonatal onset of AIDs[5–12]. These patients show some common features such as immuno deficiency, facial dysmorphisms, fever, polymorphic skin rashes, hepatosplenomegaly, neonatal-onset severe pancytopenia or dyshematopoiesis. Other unique features include macrothrombocytopenia and recurrent hemophagocytic lymphohistiocytosis (HLH) episodes in the most frequent and severe form of AID caused by the R186C substitution (NOCARH syndrome). Other variants affecting the same gene have been reported to variably affect CDC42 functional behaviour and cause a clinically hetero-geneous syndromic developmental disorder[13]. Among these, a mis-sense change (c.191 A > G, p.Y64C) located in the N-terminal part of CDC42 underlies the Takenouchi-Kosaki syndrome, which is char-acterized by intellectual delay, growth retardation, dysmorphic facial features, macrothrombocytopenia and recurrent infections[14,15]. This peculiar phenotype has also been reported in a patient with a late onset systemic inflammatory disease and myelofibrosis[16]. Although this patient presented with elevated levels of pro-inflammatory cyto-kines (IL-6, IL-18, IL-18BPa, CXCL9), C-reactive protein (CRP) and ery-throcyte sedimentation rate (ESR) compared to healthy controls, these levels were lower to those observed in NOCARH patients, suggesting a milder inflammation. Thus, a better mechanistic characterization is essential for improved therapeutic management of these life-threatening conditions.

In this work, we therefore sought to explore the mechanisms by which the CDC42 variants trigger severe AID. We show that the two disease-causing variants, R186C and *192 C*24, which are trapped in the Golgi apparatus, may exacerbate disease through endoplasmic reticulum (ER) stress, STING activation and Type-I IFN -mediated pathology.

## Results

Here, we report the further mechanistic characterization of seven unrelated patients with *CDC42* mutation-related conditions (Supple-mental Material, Supplementary Table 1). Four patients (P1-4) carry the recurrent R186C variant present in the juxta-membranous polybasic region[5,8,10]; two patients (P5-6) carry the *192 C*24 variant due to a mutation in the stop codon which adds up a stretch of 24 amino acids at the C-terminal part of CDC42(refs. 9,10), while one patient (P7) carries the Y64C mutant located in the highly mobile Switch II region of CDC42(ref. 16) (Fig. 1A, B). These variants were systematically com-pared with the other C-terminal C188Y mutant which affects the cysteine residue of the CAAX sequence that is geranyl-geranylated and consequently cannot be lipidated[10,12,17].

### CDC42 mutants exhibit a diverse subcellular localization

First, we compared the subcellular distribution of all these CDC42 variants in the THP-1 monocytic cell line using co-stainings with Golgi and nucleus markers (Fig. 1C). The degree of colocalization between CDC42 and the Golgi was quantified using the Pearson's Coefficient (Fig. 1D), as described[5–9,11]. We show here that the Y64C variant exhibits a diffuse subcellular localization similar to what is observed for the wild-type (WT) CDC42. The non-membrane anchored C188Y mutant accumulates in the nucleus. In accordance with previous reports, the R186C and *192 C*24 variants are the only ones to exhibit a retention in the Golgi due to aberrant palmitoylation[5–9,11,17]. Moreover, we demon-strate that the R186C variant is quantitatively more trapped in the Golgi compared to the *192 C*24 mutant (Fig. 1D).

### CDC42 R186C induces a block in anterograde trafficking

Because of the peculiar retention of two C-terminal CDC42 mutants in the Golgi, we next wondered whether they affect the Golgi function, and in particular the intracellular bidirectional trafficking between the endoplasmic reticulum (ER) and the Golgi.

To study the anterograde ER-to-Golgi transport, we followed the trafficking of endogenous proto-collagen of type I (PC-1) as a protein model[18]. For this purpose, fibroblasts from four healthy donors (HDs) or from Y64C and R186C patients were co-stained for Golgi, ER and nucleus markers together with PC-1 (Fig. 2A). The initial high tem-perature condition of cell culture is necessary for accumulating the majority of PC-1 in the ER at time 0 as shown by the quantification of the Pearson's Coefficient of 0.6-0.7 for all conditions (Fig. 2B). The accumulation of PC-1 is initially low in the Golgi as shown by a low Pearson's Coefficient of about 0.4 in all cells (Fig. 2C). Upon decreasing the temperature to 32 °C, PC-1 proteins are released from the ER and are expected to follow the normal secretory route through the Golgi apparatus. Quantifications of the Pearson's Coefficients after a 1-h shift in temperature show that cells from all four HDs and from the Y64C patient indeed exhibit a drop in PC-1−ER colocalization to 0.5 and an increase in PC-1−Golgi colocalization to 0.6-0.7 (Fig. 2B, C). By con-trast, cells endogenously expressing the CDC42 R186C mutant exhibit Pearson's Coefficients that remain high for the PC-1−ER colocalization and low for the PC-1−Golgi colocalization, similar to the ones observed at basal levels (time 0). This indicates that the anterograde trafficking is severely impaired in R186C patient cells.

We also quantified the total amount of PC-1 expression in all these fibroblasts. Initially, similar total levels of PC-1 were observed in all cells (Fig. 2D). However, upon 1 h of temperature shift, HD and the Y64C fibroblasts showed a 50% reduction in the expression levels of PC-1, probably due to the processing and secretion of a fraction of PC-1 proteins. In contrast, PC-1 expression levels slightly increased in fibroblasts endogenously expressing the CDC42 R186C mutant. Alto-gether, these results indicate that the Golgi-trapped CDC42 R186C variant exhibits a specific block of the anterograde ER−Golgi trafficking.

### CDC42 R186C inhibits retrograde transport

To investigate the functionality of the retrograde Golgi−ER transport in these cells, we used the Cholera toxin B subunit (CtxB) model[19] with costainings for Golgi, ER and nucleus (Fig. 3A). Upon internalization, most CtxB reaches the Golgi. For all fibroblasts, CtxB-Golgi colocaliza-tion was initially similar (Fig. 3B) and higher than the CtxB-ER colocali-zation (Fig. 3C) (Pearson's Coefficient of 0.7 *versus* 0.5). Eight hours later, the reverse was true in HD and Y64C cells, as most CtxB had reached the ER (Pearson's Coefficient: 0.7) and fewer was still present in the Golgi (Pearson's Coefficient: 0.5). Conversely, no changes were observed for the R186C patient cells indicating that the Golgi-trapped CDC42 R186C variant also impairs the retrograde Golgi−ER trafficking pathway.

### CDC42 R186C triggers ER stress and the unfolded protein response

Next, we hypothesized that the defective intracellular trafficking caused by the CDC42 R186C pathogenic mutant could increase ER stress and trigger the unfolded protein response (UPR). The molecular chaperone binding immunoglobulin protein (BiP) is considered as the main sensor in the activation of the UPR, which is a stress response triggered to restore protein and consequently cellular homoeostasis. Accordingly, we found an increased expression of BiP by microscopy (Fig. 4A) and biochemistry (Fig. 4B) techniques in baseline conditions in CDC42 R186C patient's cells but not in HD and Y64C cells. Inter-estingly, upon treatment with thapsigargin, an inhibitor of the sarco-/endoplasmic reticulum $Ca^{2+}$-ATPase (SERCA) that induces ER stress, BiP expression levels were solely increased in HD and Y64C fibroblasts (Fig. 4B), suggesting that BiP levels measured in resting R186C fibro-blasts are already maximal. Furthermore, only THP-1 cells expressing Golgi-trapped R186C and *192 C*24 CDC42 variants exhibited an excess of BiP expression upon thapsigargin treatment (Fig. 4C).

Next, we used treatment with thapsigargin and qRT-PCR evalua-tion of 3 UPR-dependent genes (*HSPA5*, *ATF4* and *DDIT3*) in cells

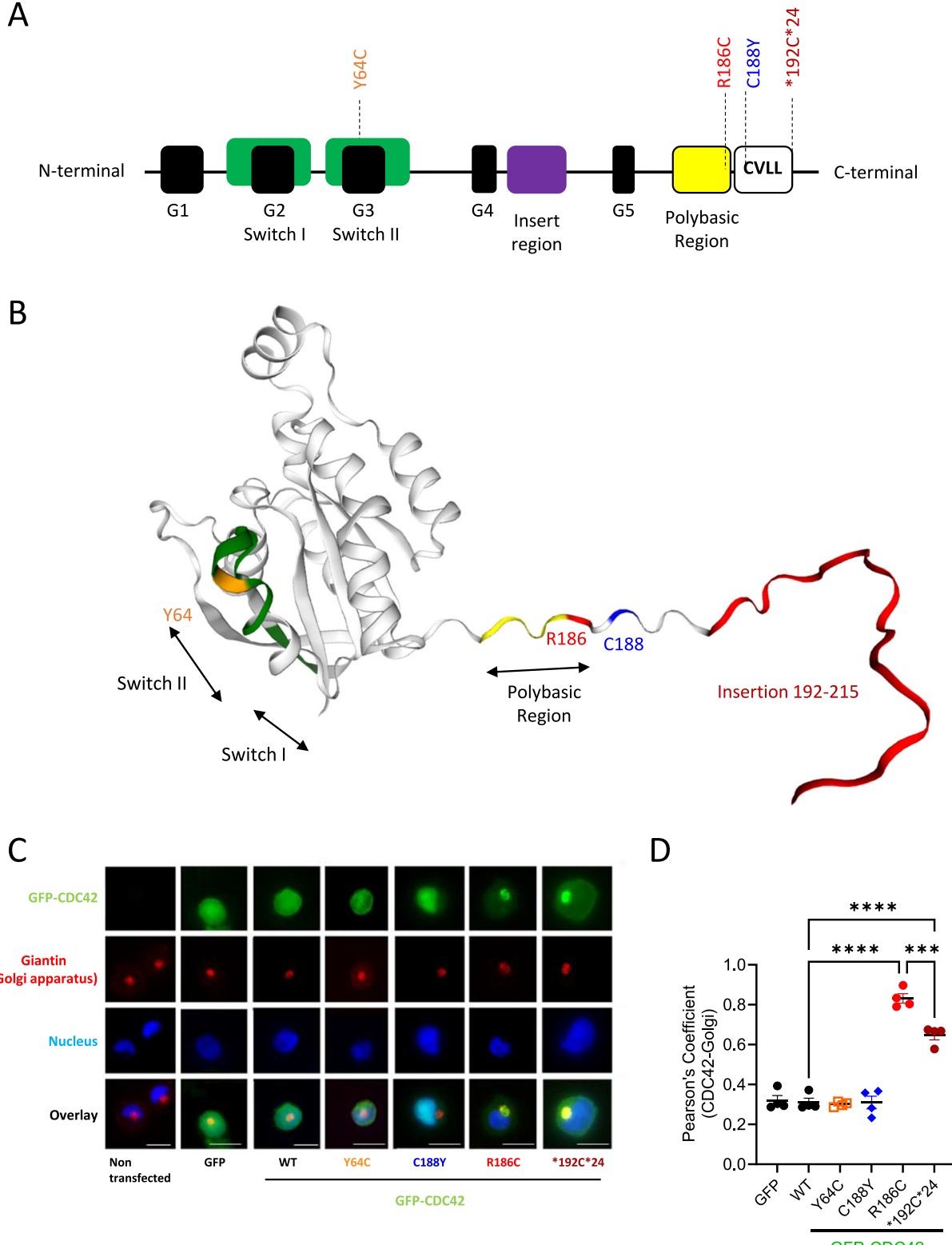

**Fig. 1 | Localization of CDC42 variants identified in patients. A** Organization of CDC42 domains and indication of the positions of the pathogenic variants under study. **B** Three-dimensional structure of CDC42 showing the CDC42 mutations of amino acids Y64, R186 and C188, and the insertion of the 24 amino acids at the C terminus highlighted in orange, red, blue, and dark red, respectively. The switch regions are indicated by black arrows and green colour, and the polybasic region is represented in yellow. **C** Subcellular localization of GFP-CDC42 variants in THP-1 cells co-stained for the Golgi and nucleus. Scale bars: 10 μm. **D** Quantification of the degree of Golgi−CDC42 co-localization for different variants using the Pearson's Coefficient. One dot represents the mean value from about 15 cells from one independent experiment. Results are shown as means +/- SEM of four biological replicates ($n = 4$) and the significance levels were calculated using ordinary one-way ANOVA (***$P = 0.0003$; ****$P < 0.0001$). Source data are provided as a Source Data file.

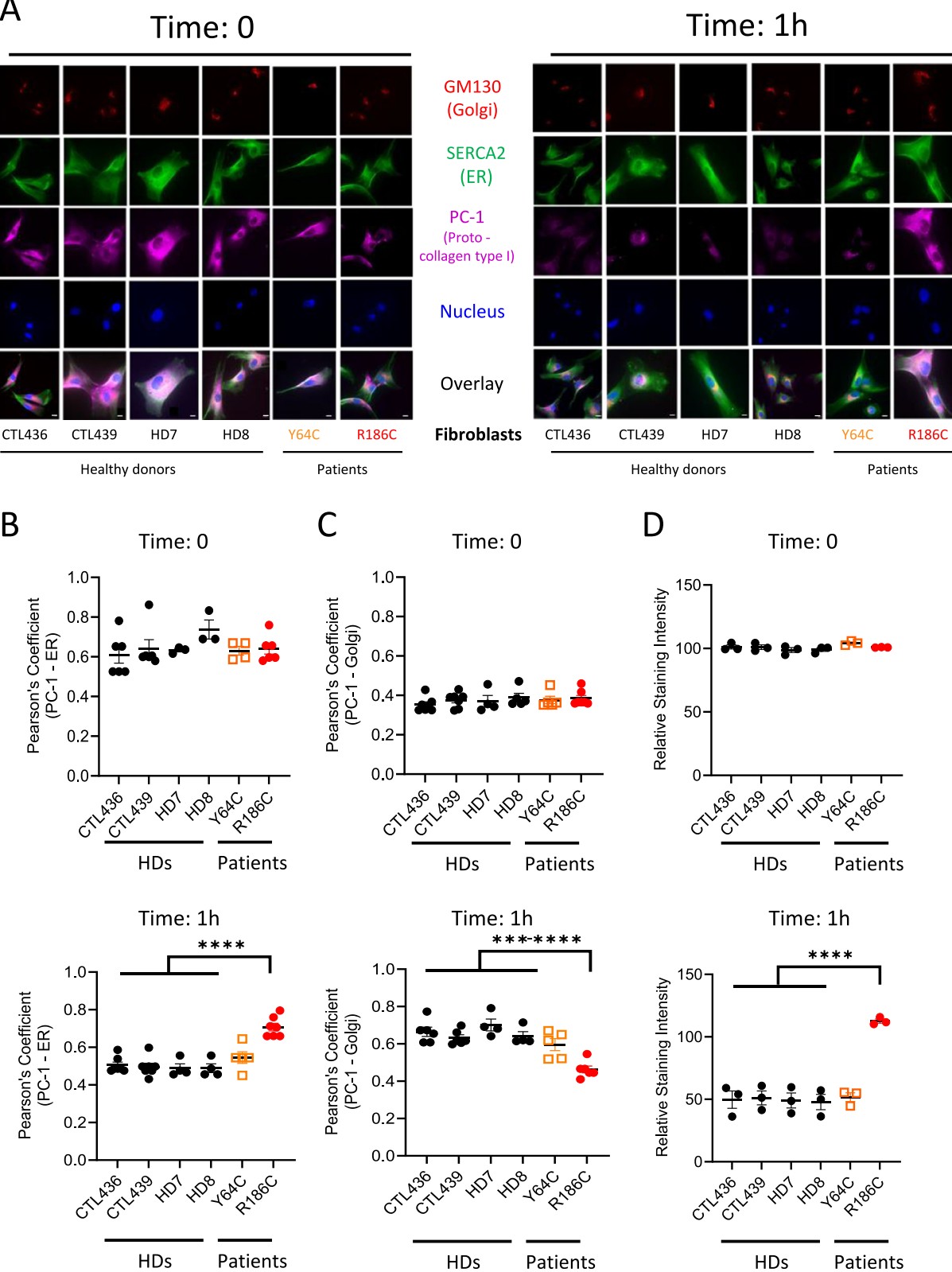

**Fig. 2 | Impact of CDC42 variants on anterograde transport.**
**A** Immunofluorescence analyses of PC-1, ER, Golgi and nucleus localizations in 4 different healthy donor (HDs) fibroblasts, patients Y64C and R186C at 0 and 1 h. Images are representative of at least 3 independent experiments. Scale bars: 10 μm. Quantifications of the colocalizations between PC-1 and the ER (**B**) or the Golgi (**C**) at the indicated times for each cell type. **D** Analysis of total PC-1 fluorescence intensity following anterograde transport. The relative intensity of PC-1 expression was normalized to 100 from the mean values of the 4 HDs at time 0. In all graphs, one dot represents the mean value from about 15 cells from one independent experiment. Results are shown as means +/- SEM of at least three biological replicates and the significance levels were calculated using ordinary one-way ANOVA (***$P = 0.0006$, ****$P < 0.0001$). Source data are provided as a Source Data file.

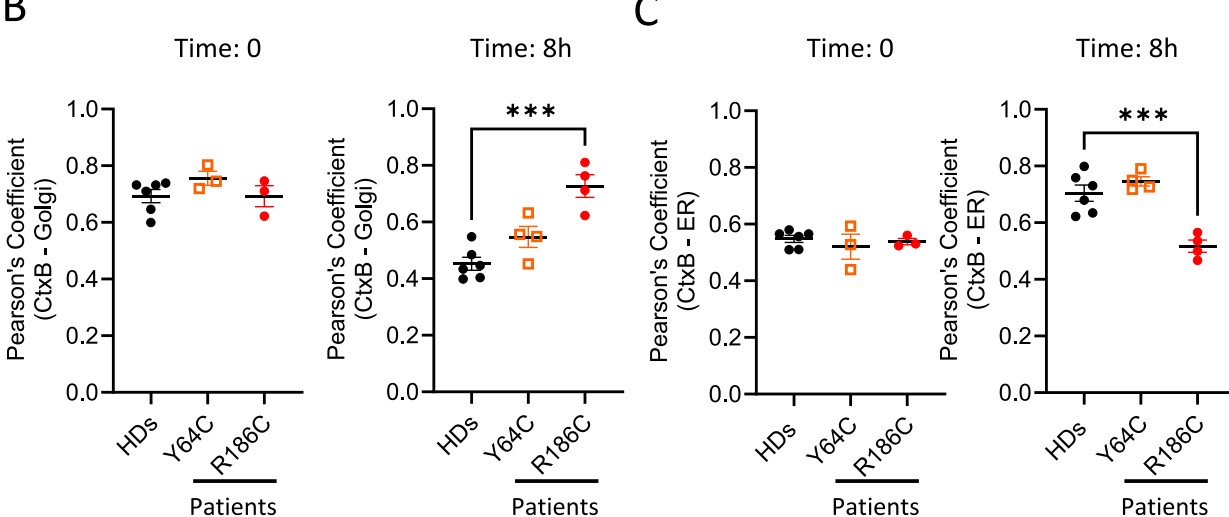

**Fig. 3 | Impact of CDC42 variants on retrograde transport.**
**A** Immunofluorescence analyses of retrograde transport assay of Cholera toxin B subunit (CtxB) in fibroblasts from the HD8 healthy donor and from the Y64C and R186C CDC42 patients at times 0 and 8 h. Co-stainings for Golgi, ER and nuclei were performed. Images are representative of at least 3 independent experiments. Scale bars: 10 μm. Quantifications of microscopy images are shown for the CtxB−Golgi

(**B**) and CtxB−ER (**C**) colocalizations at the indicated times. Transport assay data from 4 healthy donors were pooled. In all graphs, one dot represents the mean value from about 15 cells from one independent experiment. Results are shown as means +/- SEM of at least three biological replicates and the significance levels were calculated using ordinary one-way ANOVA (***$P < 0.007$). Source data are provided as a Source Data file.

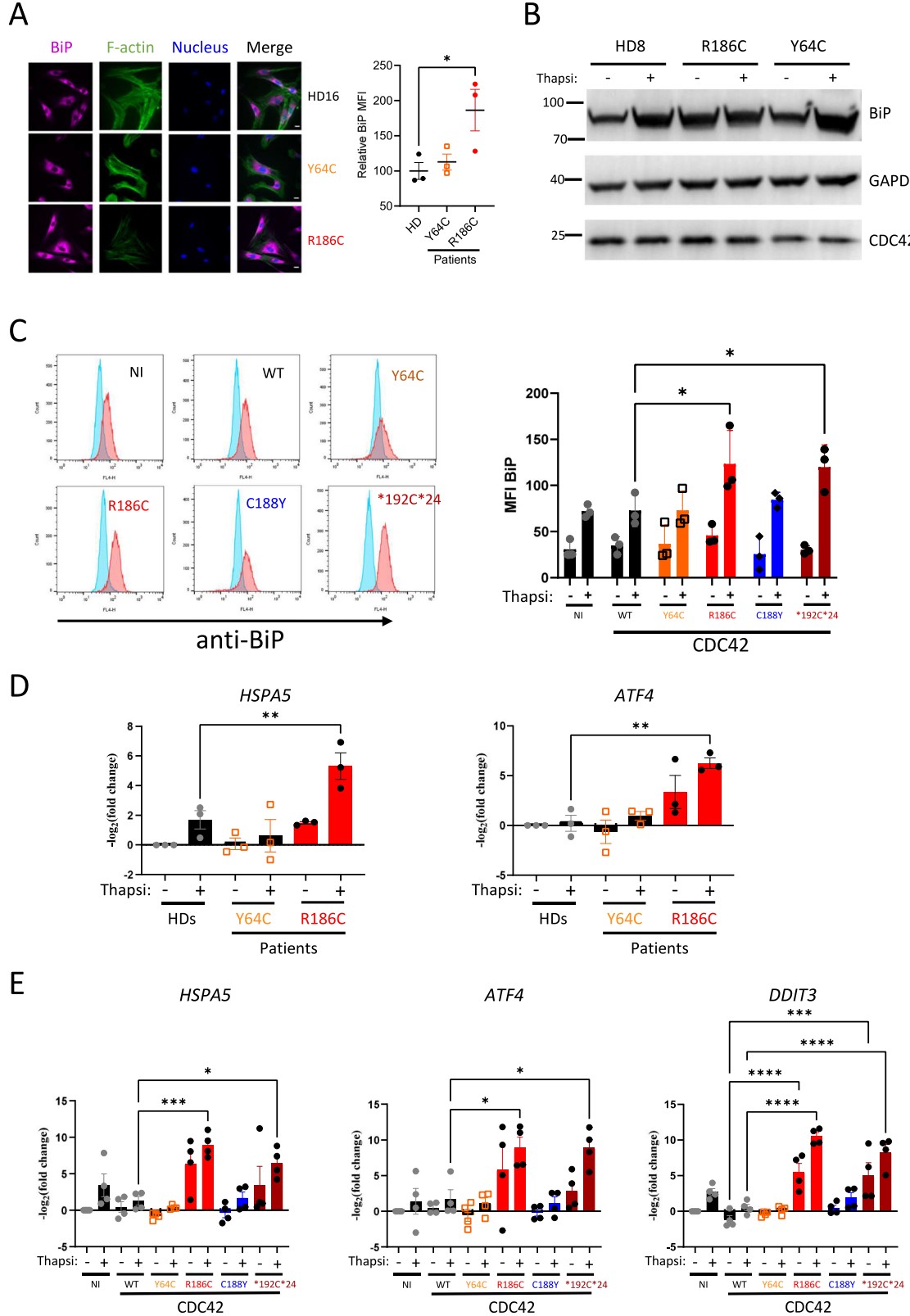

expressing CDC42 variants. In our hands, DDIT3 mRNA was not expressed in fibroblasts. However, both HSPA5 and ATF4 mRNA were strongly overexpressed in CDC42 R186C patients' fibroblasts compared to HDs upon thapsigargin treatment (Fig. 4D). CDC42 Y64C fibroblasts behaved like HDs cells. Overexpression of these variants in THP-1 cells confirmed the findings obtained in patients-derived fibroblasts (Fig. 4E). Only the Golgi-trapped R186C and *192 C*24 CDC42

variants showed marked increases in all tested UPR-dependent genes upon thapsigargin treatment. DDIT3 mRNA levels were already increased at the basal level specifically in the two Golgi-localised variants. Thus, the impairments in both anterograde and retrograde trafficking observed in cells expressing the Golgi-trapped CDC42 variants are likely to be responsible for ER stress which has been shown to be related to inflammation[20,21].

**Fig. 4 | Impact of CDC42 variants on ER stress. A** Immunofluorescence analyses of ER stress in fibroblasts from the HD16 healthy donor and from the Y64C and R186C CDC42 patients. Left: co-stainings for BiP, F-actin and nuclei are shown. Images are representative of 3 independent experiments. Scale bars: 10 μm. Right: quantification of microscopy images. One dot represents the mean value from about 15 cells from one independent experiment. Results are shown as means +/- SEM of three biological replicates and the significance levels were calculated using ordinary one-way ANOVA (*: $P = 0.0358$). **B** Western blot analysis of BiP and CDC42 expression in HD8, Y64C and R186C fibroblasts treated with DMSO (vehicle, -) or thapsigargin (+). GAPDH is shown as a loading control. The Molecular Weights (kDa) are indicated. **C**, Flow cytometry analyses of BiP expression in THP-1 cells expressing WT or mutant CDC42. Left: examples of BiP expression FACS profiles in cells treated with DMSO (vehicle; turquoise) or thapsigargin (red). Right: quantifications of the BiP Mean Fluorescence Intensity (MFI) in cells treated with DMSO (-) or thapsigargin (+). One dot represents the MFI from one independent experiment. Results are shown as means +/- SEM from three biological replicates and the significance levels were calculated using two-way ANOVA (*$P < 0.0359$). **D** Expression levels of HSPA5 (left) and ATF4 (right) mRNA in healthy donors (HDs) or CDC42 patients' fibroblasts treated with DMSO (-) or thapsigargin (+). **E** Expression levels of HSPA5 (left), ATF4 (center), and DDIT3 (right) mRNA in THP-1 cells expressing WT or variants CDC42 upon DMSO (-) or thapsigargin (+) treatment. NI: non infected. For panels D and E, each dot represents the mean value of three technical replicates from one independent experiment. Results are shown as means +/- SEM from at least three biological replicates and the significance levels were calculated using ordinary one-way ANOVA (*$P < 0.0328$; **$P < 0.0081$; ***$P < 0.0004$; ****$P < 0.0001$). Source data are provided as a Source Data file.

## Golgi-trapped CDC42 mutants induce STING hyperactivation

As previously described, the small pool of CDC42 naturally present in the Golgi apparatus in physiological conditions can interact with the COPI complex through coat protein complex subunit γ 1 (COPγ1) and regulate COPI-dependent bidirectional transport[22–24]. Here, we have shown that an excess of pathogenic CDC42 in the Golgi blocks the bidirectional ER-Golgi route. Interestingly, we and others have shown that this particular defect is also observed in patients with heterozygous mutations in *COPA*[21,25–28], encoding for one of the subunits of the COPI complex. As these COPA patients have been shown to develop inflammatory syndromes related to alterations in STING trafficking, we next studied whether Golgi-trapped CDC42 variants also exhibit changes in the subcellular localization of STING.

To this aim, we stained patients' and HDs fibroblasts for STING, Golgi and nucleus. We observed STING enrichment in the Golgi from the R186C patient fibroblasts but not in Y64C or in any four HDs cells (Fig. 5A). Furthermore, in accordance with this finding, ectopic over-expression of WT, Y64C and C188Y CDC42 in THP-1 cells did not impact STING localization (Fig. 5B). By contrast, the two Golgi-trapped variants R186C and *192 C*24 elicited STING accumulation in the Golgi, with higher efficiency for R186C compared to *192 C*24. This enrichment of STING was abrogated when these mutants were precluded from interacting with the COPγ1 subunit of COPI by mutating the K183K184 motif in CDC42(ref. 22) (Fig. 5C). These data point to the crucial interaction of CDC42 with COPI for intracellular transport. Thus, although the CDC42 K183S/K184S/R186C and K183S/K184S/*192 C*24 mutants retain a strong Golgi accumulation (Fig. 5D), only CDC42 R186C and *192 C*24 trigger STING accumulation in the Golgi, and this effect depends on the interaction of CDC42 with COPI.

Because STING is only phosphorylated and active when it reaches the Golgi[29–34], we next monitored several downstream readouts / effectors of the STING pathway. Phosphorylation of IRF3 (P-IRF3) was specifically increased only in THP-1 cells expressing the R186C and *192 C*24 variants (Fig. 6A). R186C patients' cells also showed higher levels of P-IRF3 (Fig. 6B) compared to 4 HDs and the Y64C cells. The basal level of STAT1 phosphorylation was very weakly increased in R186C fibroblasts (Fig. 6C). However, STAT1 phosphorylation was markedly exacerbated in THP-1 cells expressing the R186C and *192 C*24 CDC42 variants, but not the Y64C and C188Y variants, upon IFNα stimulation (Fig. 6D).

Altogether, these results indicate that the STING signalling pathway is hyperactivated in cells expressing the Golgi-trapped CDC42 variants.

## Golgi-trapped CDC42 patients show increased levels of IFNα at the onset of the disease

To test the possibility that the hyperactivation of the STING pathway elicits an increased type I IFN (IFN-I) response, we first measured the levels of IFNα present in available serum or plasma samples from patients carrying Golgi-trapped CDC42 mutations at different stages (onset, prior to bone marrow transplantation, late) and under different treatments (Fig. 7; Supplemental Material; Supplementary Table 1). R186C patients P1-4 all had very high levels of IFNα at the onset of the disease or upon early treatment. These amounts of IFNα were reduced to various extent upon diverse treatments or upon bone marrow transplantation, although basal levels as in HD were not systematically reached. *192 C*24 Patient 5 also had very high levels of IFNα at an early time point which was resolved later on. *192 C*24 Patient 6 was on treatment with various doses of corticosteroids due to recurrent vasculitis. This may explain why her IFNα levels were systematically low in the measured samples (Fig. 7).

## Golgi-trapped CDC42 mutants induce high expression of IFN-stimulated genes (ISG)

Next, we measured the mRNA expression of six IFN-stimulated genes (ISG) to calculate the type I IFN score. Contrary to Y64C patients' cells, the R186C cells showed a marked increase in IFN score (Fig. 8A). This was also confirmed in THP-1 cells where only the two Golgi-trapped variants showed a high type I IFN score, in contrast to the Y64C and C188Y mutants (Fig. 8B). Furthermore, although the IFN score was slightly reduced in cGAS KO THP-1 cells (Fig. 8C, D), R186C and *192 C*24 expressions were still able to elicit higher ISG production compared to WT CDC42. However, the increased ISG expression in R186C and *192 C*24 was dependent on STING as it was completely abrogated in STING KO THP-1 cells (Fig. 8E). This indicates a strict requirement for STING to induce ISG expression by these two Golgi-trapped CDC42 variants.

## Discussion

Overall, we extend the spectrum of cellular dysfunction previously observed in Golgi-trapped CDC42 patients[5–12,17]. More specifically, we provide additional mechanistic insights for the autoinflammatory phenotype caused by CDC42 R186C and *192 C*24 variants.

Indeed, we show that the recurrent CDC42 R186C mutant, reported now in 13 patients affected by autoinflammation, impairs anterograde and retrograde trafficking between the ER and the Golgi, and elicits ER stress. This deficient intracellular trafficking in turn leads to STING accumulation in the Golgi as we and others previously reported for patients carrying mutations in genes involved in the transport machinery such as *COPA*[21,25–27] and, more recently, *ARF1*(ref. 35). Importantly, we demonstrate that the interaction between COPI and the Golgi-trapped CDC42 mutants is essential for triggering STING enrichment in the Golgi. Of note, the R186C mutant is more potent than the *192 C*24 one in inducing the functional defects described above, and this correlates with increased trapping of R186C in the Golgi compared to the *192 C*24 mutant. Consequently, only these two Golgi-localized CDC42 mutants induce STING activation and ISG expression. In our hands, the other C-terminal C188Y mutant, which is not trapped in the Golgi, does not induce STING accumulation in the Golgi, nor STING activation, nor ISG expression. Our data also suggest that ISG expression, induced by both Golgi-trapped CDC42 mutants, is largely cGAS-independent which seems at odds with the recent report of defective

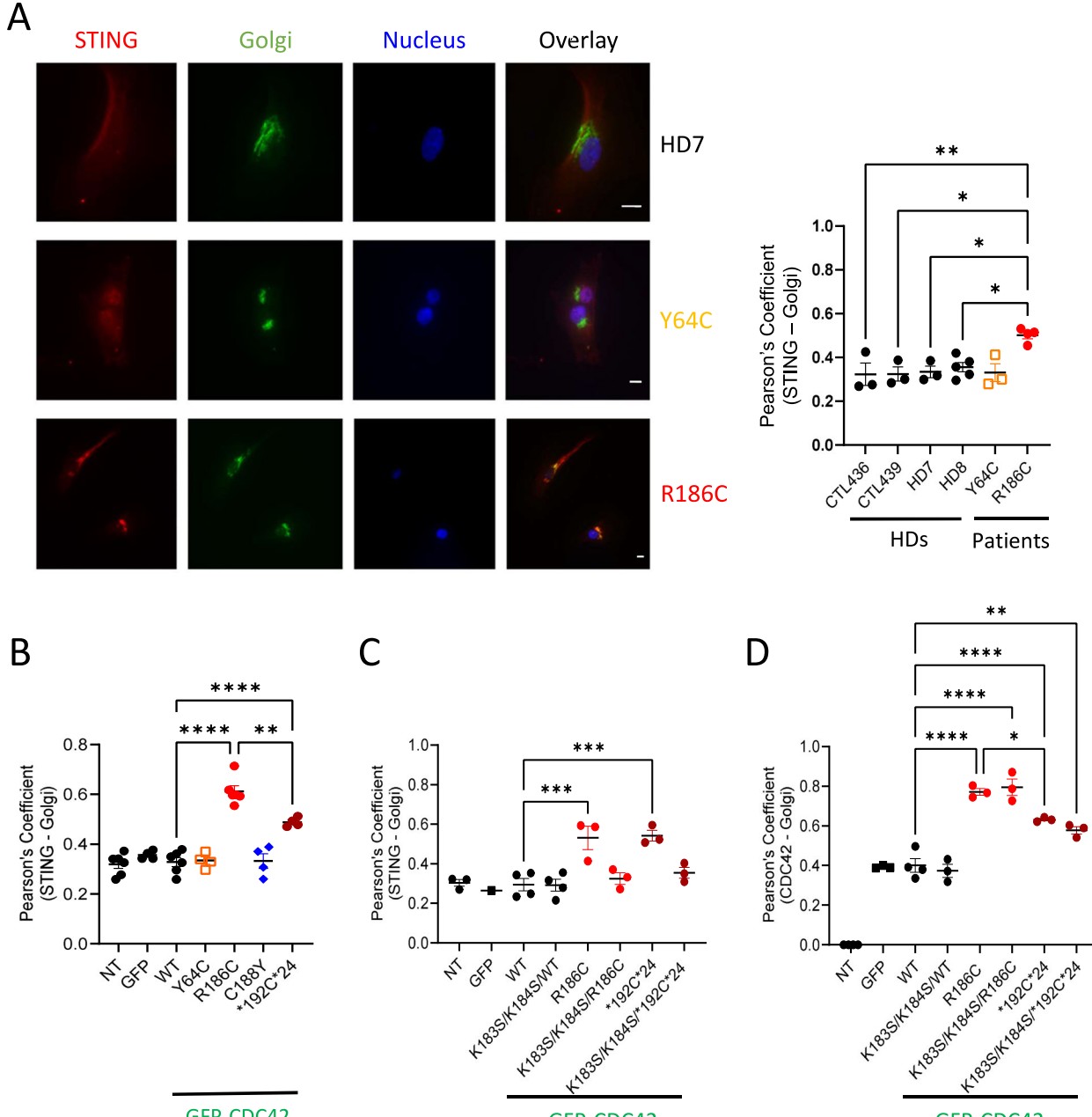

**Fig. 5 | Golgi-trapped CDC42 variants induce STING accumulation in the Golgi in a COPI-dependent manner. A** Left: Subcellular stainings of STING, Golgi and nuclei in control and patients' fibroblasts. Scale bars: 10 μm. Right: Quantification of the degree of STING−Golgi co-localization in each condition. **B** Measurement of STING−Golgi co-localization in THP-1 cells expressing different forms of CDC42. Quantifications of the degrees of STING−Golgi (**C**) or CDC42−Golgi (**D**) co-localizations in THP-1 cells expressing different mutants of CDC42, including some with the K183S/K184S double mutation which inhibits COPI binding. NT: non transfected. In all the graphs, each dot represents the mean value from about 15 cells from one independent experiment. Results are shown as means +/- SEM from at least three biological replicates and the significance levels were calculated using ordinary one-way ANOVA (*$P < 0.0192$; **$P < 0.0098$; ***$P < 0.0006$; ****$P < 0,0001$). Source data are provided as a Source Data file.

mitochondria releasing DNA in the cytosol of R186C and C188Y fibroblasts[12]. Potentially, this impairment of the mitochondria structures is only present in CDC42 patients' fibroblasts but not in THP-1 cells expressing pathogenic CDC42 variants. Using STING-deficient cells, our data support an absolute requirement for STING expression for inducing ISG expression specifically by Golgi-trapped CDC42 mutants. Thus, there seems to be several possible mechanisms to hyperactivate the STING pathway in these pathological conditions. Finally, the Y64C mutant does not exhibit any defects in the readouts we studied, probably in line with the late onset of the disease for this patient[16]. Potentially, other mechanisms are at play in this variant.

ER stress plays a complex role in the induction of inflammation[20]. Defects in the retrograde Golgi-to-ER transport caused by mutations in different COPI subunit proteins, were previously described to result in an increased ER stress[21,36–39]. Similarly, we observed an increased ER stress measured by BiP expression and UPR-dependent genes specifically in CDC42 R186C and *192 C*24 variants. One can assume that the increased ER stress, that we have observed here, could result from an impaired retrieval of chaperone proteins to the ER, due to a strongly reduced Golgi-to-ER retrograde transport. In addition, accumulation of nascent proteins in the ER, due to impairment in ER-to-Golgi ante-rograde transport induced by CDC42 R186C, could also be responsible

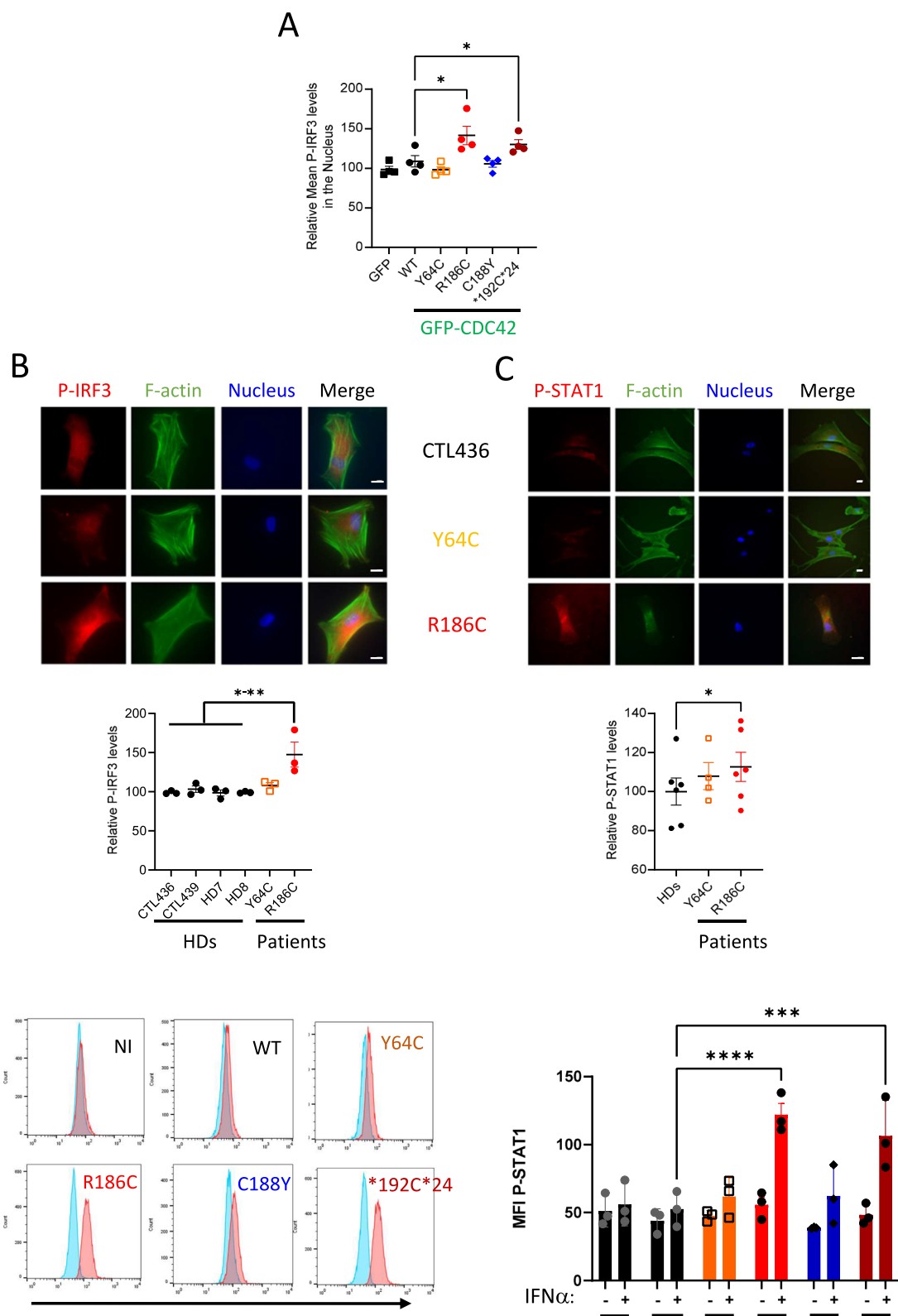

**Fig. 6 | Golgi-trapped CDC42 variants induce STING pathway activation.**
**A** Measurement of P-IRF3 intensity in the nuclei of THP-1 cells expressing GFP-CDC42 variants. Immunocytochemistry stainings and quantifications of P-IRF3 (**B**) and P-STAT1 (**C**) intensities in healthy donors (HDs) or CDC42 patients' fibroblasts. Scale bars: 10 μm. In all graphs, one dot represents the mean value of fluorescence intensities from about 15 cells from one independent experiment. **D** Flow cytometry analyses of P-STAT1 expression in THP-1 cells expressing WT or mutant CDC42. Left: examples of P-STAT1 expression profiles in cells non stimulated (turquoise) or stimulated with IFNα (red). Right: quantifications of the P-STAT1 Mean Fluorescence Intensity (MFI) in cells upon IFNα stimulation. Each dot represents the MFI from one independent experiment. For all graphs, results are shown as means +/- SEM of at least three biological replicates and the significance levels were calculated using ordinary one-way ANOVA (*$P < 0.0424$; **$P < 0.0086$; ***$P = 0.0004$; ****$P < 0.0001$). Source data are provided as a Source Data file.

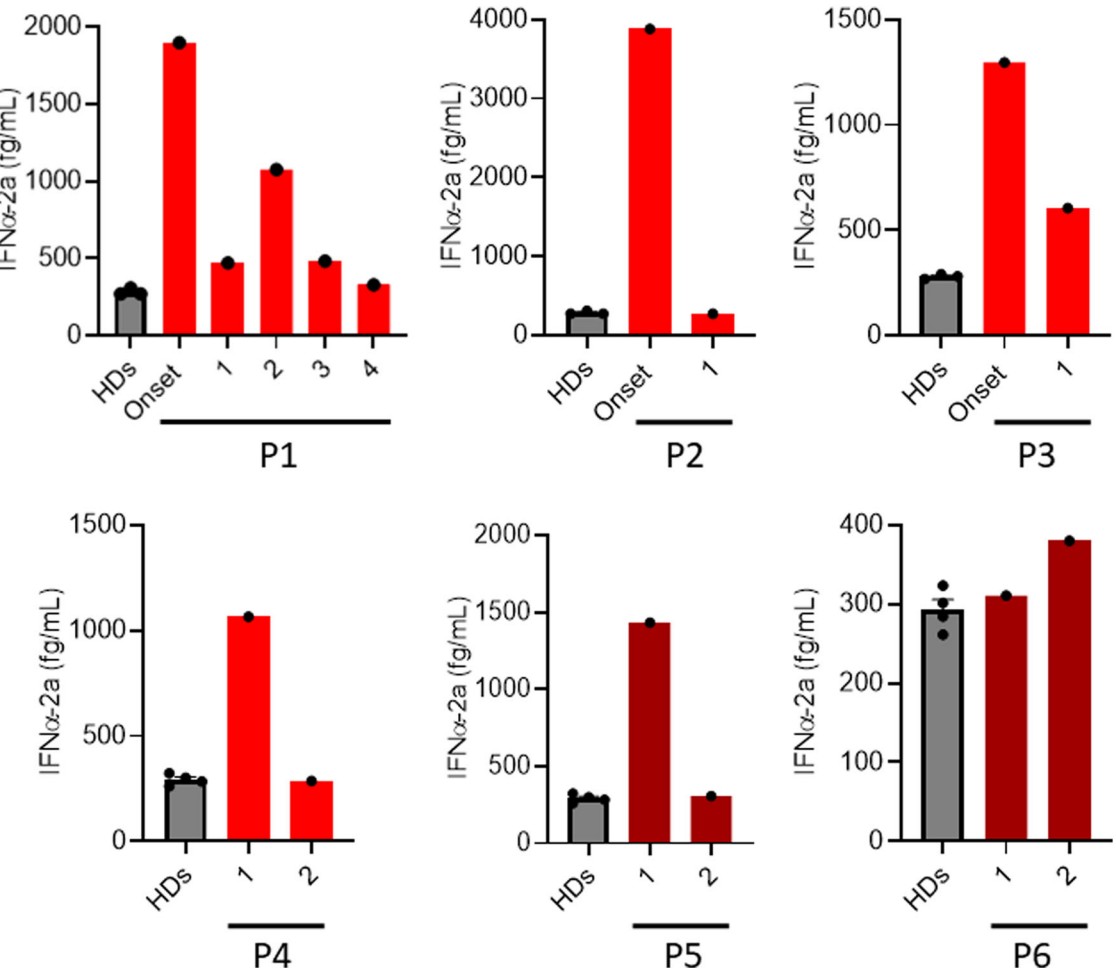

**Fig. 7 | Golgi-trapped CDC42 variants are associated with high IFNα levels in the blood.** Quantification of IFNα in sera or plasma from R186C (P1-4) and *192 C*24 (P5-6) CDC42 patients at the onset of the disease or after different treatments: P1 → 1: probable virosis (ongoing glucocorticoids and anakinra), 2: infection by adenovirus and rhinovirus (ongoing glucocorticoids, anakinra and cyclosporine), 3: ongoing cyclosporine, glucocorticoids and anakinra, 4: bone marrow transplantation. P2 → 1: high dose of glucocorticoids and anakinra. P3 → Onset: improved state. 1: Re-inflammatory state, anti-TNF treatment. P4 → 1: anakinra, 2: glucocorticoids and canakinumab. P5 → 1: anakinra, 2: high dose of anakinra. P6 → 1 and 2: corticosteroids and intravenous Ig (Immunoglobulin response therapy). Each dot represents the mean value of three technical replicates from one independent experiment. HDs measurements were performed in three or four biological replicates and the results are shown as means +/- SEM. Source data are provided as a Source Data file.

for ER stress. Increased ER stress together with defective bidirectional trafficking are therefore key pathogenic mechanisms caused by the CDC42 R186C variant responsible for the NOCARH syndrome.

Thus, next to increased NF-κB[11] and Pyrin[8,40] activation, high ER stress and STING hyperactivation are here unveiled as additional proinflammatory pathways elicited by the pathogenic CDC42 Golgi-trapped variants in both patients' cells and upon ectopic expression. In agreement with the hyperactivation of the STING pathway, we show marked increases in IFNα concentration for all five unrelated R186C and *192 C*24 CDC42 patients for whom blood samples at the onset of the disease or upon early treatment were available. These high IFNα levels were reduced to normal upon transplantation, indicating that mutated immune cells are responsible for the observed defects.

In this way, our results may lead to novel insights in treatment approaches, by targeting the STING−Type I IFN pathway in these patients. As far as we know, the H-151 STING inhibitor largely used in vitro is not used in clinics. However, several STING inhibitors are in pre-clinical trials. The mainstay of treatment for decreasing the activation of the STING−Type I IFN pathway has been to use JAK inhibitors such as ruxolitinib or baricitinib, but they block the signalling of many cytokine receptors, lacking specificity[12,41–44]. More precise inhibitors

targeting the STING−Type I IFN pathway have been developed recently, including Anifrolumab, an anti- type I IFN receptor subunit 1 (IFNAR1) monoclonal antibody, which seems to provide better efficacy[45,46]. Finally, STING degradation approaches[47] could also be valuable for treating patients presenting with a hyperactivation of the STING−Type I IFN pathway. Based on our data, these therapies could be of benefit for the patients harbouring Golgi-trapped CDC42 variants.

Altogether, our findings showcase a layer of complexity regarding the extreme variability in immunological and cellular phenotypes provoked by pathogenic variants in a single gene despite all patients being affected by autoinflammation.

## Methods

### Study oversight
Human studies were carried out according to French law on biomedical research and to the principles outlined in the 1975 Helsinki Declaration and its modification. Institutional review board approvals were obtained (DC-2023-5921 and IE-2023-3004 from the French Ministry of Higher Education and Research). All patients provided written informed consent for the conservation and use of their blood

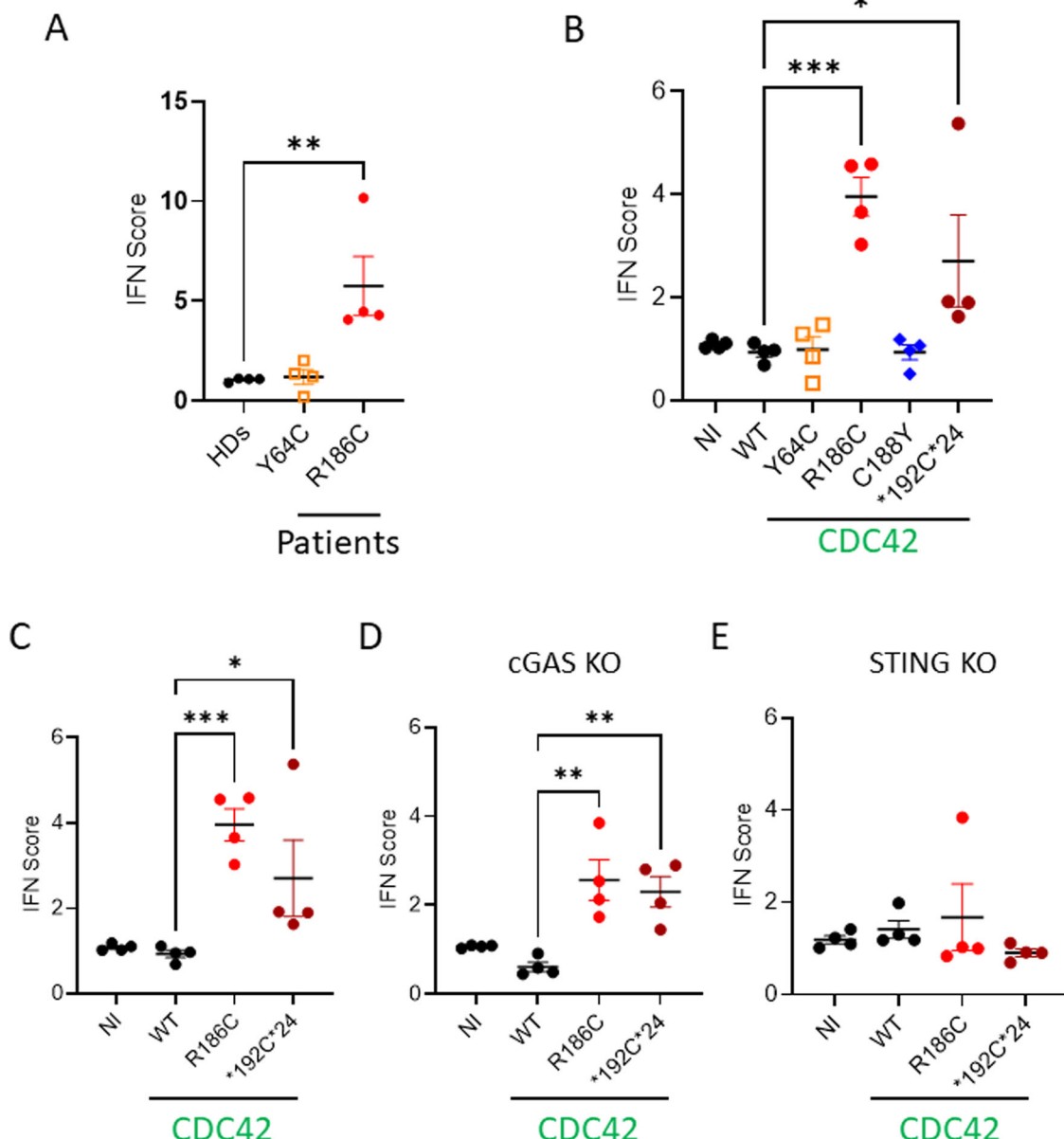

**Fig. 8 | Golgi-trapped CDC42 variants induce increased Interferon signature gene expression in a STING-dependent way.** IFN scores (ISG: *IFI27*, *IFI44L*, *IFIT1*, *ISG15*, *RSAD2* and *SIGLEC1*) from 4 independent experiments were quantified in patients' fibroblasts (**A**), THP-1 (**B**, **C**), THP-1 cGAS KO (**D**) and THP-1 STING KO (**E**) cells expressing different CDC42 mutants. Panels B and C are partly duplicated. NI: non-infected. For each gene, an expression mean value was obtained from three technical replicates. An IFN score was then obtained from the median values for the 6 ISG and corresponds to one dot from one independent experiment. Results are shown as means +/- SEM from four biological replicates and the significance levels were calculated using ordinary one-way ANOVA (*$P$ < 0.0295; **$P$ < 0.0080; ***$P$ < 0.0003). Source data are provided as a Source Data file.

samples and cells for research. Patients were included in our study based on the CDC42 mutation they carry. Thus, sex and/or gender analysis was not carried out.

## In silico CDC42 structural analysis

The SWISS-MODEL server (https://swissmodel.expasy.org/) was used based on the template of the three-dimensional X-ray structure of CDC42 (PDB accession number: 5c2j.1.B). It shows the well-known structure of CDC42 and predicts the additional 24 amino acids to the protein's C terminus present in the *192 C*24 mutant. This additional sequence is predicted to be an unstructured region with a low confidence level.

## Plasmid constructs

The GFP-CDC42 plasmid previously described[48] encodes for the ubiquitous isoform 1 of CDC42 and contains a GFP tag in N-terminal. From this construct, R186C, K183S/K184S/R186C, C188Y and Y64C were obtained by site-directed mutagenesis (Quick change kit, Agilent Technologies). The *192 C*24 and K183S/K184S/*192 C*24 plasmids were produced by Thermo Fisher. The pmaxGFP vector was provided by Lonza. Plasmids encoding for Gag, Pol and VSV-G were provided by Thomas Henry (*Centre International de Recherche en Infectiologie*, Lyon, France). Nucleotide coding sequences of myc-tagged CDC42 WT and variants cloned into the pLenti-EF1a-IRES-EGFP (Creative Biolabs) were provided by the CIGEx facility (CEA, Fontenay-aux-Roses, France).

## Cells

Primary fibroblasts from the Y64C patient (P7)[16] and healthy donors[21] were obtained from the Leuven biobank of primary immunodeficiency. Primary fibroblasts from the R186C patient and healthy donors were previously described[11]. Fibroblasts and HEK-293T (ATCC # CRL-11268) cells were cultured in DMEM medium (Gibco # 31966021) supplemented with 10% Fetal Calf Serum, antibiotics (Penicillin and Streptomycin, Eurobio # CABPES01-0U) and sodium pyruvate (Gibco # AA360039). The human THP-1 (ATCC # TIB-202) monocytic cell line was provided by Serge Benichou (*Institut Cochin*, Paris). THP-1 cGAS KO and STING KO were kindly provided by Marie-Louise Frémond (*Institut Imagine*, Paris). THP-1 cells were cultured in RPMI (Gibco # 61870010) medium supplemented with 10% Fetal Calf Serum (Gibco # A5256801), antibiotics (Penicillin and Streptomycin) and sodium pyruvate. All cells were regularly tested for Mycoplasma (Lonza).

## Patients samples

For serum extraction, blood was collected in an SST tube, spun at 3000 rcf for 10 min and stored at −20 °C. For plasma extraction, blood was collected in EDTA tubes, spun twice at 2000 rcf for 10 and 15 min. Sera and plasmas from healthy donors used as controls were collected from age and sex-matched individuals.

## Transfections

$2 \times 10^6$ THP-1 cells were centrifuged and washed in PBS (Gibco # 14190-094). They were then transfected by nucleofection with 3 µg of plasmid DNA in 100 µL of Cell Line Nucleofector Solution V (Lonza) using the V-001 program (Amaxa Biosystems # VCA-1003). After transfection, 500 µL of warmed and complete RPMI medium was added to the cells, which were then plated into 6-well plates containing 1.5 mL of the same medium. The plates were then incubated overnight.

## Flow cytometry

To evaluate BiP expression, THP-1 cells were treated for 20 h with DMSO (vehicle) or 10 µM thapsigargin. To assess STAT1 phosphorylation, cells were stimulated overnight with 100 U/mL IFNα1 (Sigma # SRP4569). In both experiments, cells were fixed, permeabilized, and stained as described below. Cells were first gated based on the same GFP expression levels, and then the expression of the marker of interest was checked.

The gating strategy used in transiently transfected THP-1 cells is shown in Supplementary Fig. 1A. For each experiment, the same GFP⁺ gate was used for all THP-1 cells transiently expressing GFP alone, WT or variant CDC42 (Supplementary Fig. 2A, B).

## Lentiviral transduction

$5.10^6$ HEK-293T cells were seeded in 10 cm-diameter Petri dishes in complete DMEM and left overnight at 37 °C with 5% $CO_2$. 2 h before the transfection, the medium was replaced and the following mix was prepared: Plasmid of interest (20 µg), pPAX 8.91 gag-pol expressor (10 µg), pMDG VSV-G expressor (5 µg), 1 M $CaCl_2$ and $H_2O$ (qsp 0.5 mL). This mix was added to 0.5 mL of 2X HBS, incubated at room temperature for 30 min and added overnight to the dishes. The next day, the medium was removed and replaced with 5 mL of DMEM. 24 h later, the media were recovered, centrifuged at 1000 g for 5 min, and filtered with a 0.45 µm filter. After addition of a sucrose solution (200 g/L sucrose, 100 mM NaCl, 20 mM HEPES, 1 mM EDTA), the samples were centrifuged for 2 h at 21.109 rpm (82.700 g) at 4 °C. The supernatant was removed and the pellet was resuspended in 100 µL of PBS 1X without $Ca^{2+}$ and $Mg^{2+}$ and incubated at 4 °C for 2 h. Lentiviruses were then stored at −80 °C. $8 \times 10^4$ THP-1 cells were transduced with viruses, and analysed by flow cytometry (BD FACSCalibur) at day 4 after infection. Infected cells were identified by GFP expression and sorted by flow cytometry (FACS ARIA III, BD biosciences) (Supplementary

Fig. 1B). Matched GFP and Myc expression was checked for all CDC42 variants (Supplementary Fig. 2C–E).

## Trafficking assays

Fibroblasts were first seeded on Lab-Teks (Chamber Slide system, Thermo Fisher # 154534).

For testing the anterograde transport, we favoured PC-1 retention in the ER by incubating the cells for 3 hrs at 40 °C in complete DMEM[18]. Cells were then shifted to 32 °C for 1 hr.

For the evaluation of the retrograde transport, cells were incubated with 0.15 µg/mL AF555-labelled CtxB (Molecular Probes, C34776, Lot number 2442122) on ice for 30 min[19]. After washing twice with PBS, the cells were incubated with a pre-warmed DMEM medium for the indicated times at 37 °C. Time 0 corresponds to the maximal retention of CtxB in the Golgi and is obtained after 2 hrs of incubation. CtxB maximally reaches the ER 8 hrs later.

## Intracellular fluorescent stainings

Cells were fixed with 4% paraformaldehyde (Electron Microscopy Sciences) for 10 min for all experiments except for P-STAT1 staining that used a cold 90%−methanol solution in ice for 15 min. Then, cells were washed once in saturation buffer [PBS 1% BSA (Sigma)], permeabilized with 0.1% Triton X-100 buffer for 10 min and washed twice in permeabilization buffer (PBS containing 0.1% saponin (Fluka Biochemika) and 0.2% BSA). Cells were then incubated for 45 min with the primary antibodies (4 µg/mL) based on the type of experiment (Supplementary Table 3). They were then washed twice with 1.5 mL of saponin buffer and incubated with secondary antibodies (2 µg/mL) (Supplementary Table 3) for 30 min in the dark. Cells were washed twice in saponin buffer and twice with PBS. Nuclei labelling was next performed with 1 µg/mL of Hoechst for 10 min in the dark. Cells were finally washed twice with 1.5 mL PBS, and mounted with VECTASHIELD (Vector Laboratories H-1700).

To evaluate the ER stress, THP-1 cells expressing WT or mutant CDC42 were treated with DMSO or 10 µM of thapsigargin (Sigma # T9033) for 20 hrs, fixed and permeabilized as we described previously[21], then incubated for 45 min with anti-GRP78 BiP antibody. F-actin was labelled with phalloidin Alexa-Fluor488.

## Meso scale discovery multiplex assay

S-PLEX human Interferon KIT (K151P3S-1, lot number Z00S0023, for IFNα-2a) was purchased from Mesoscale Discovery (MSD). S-PLEX plates were coated with linkers and biotinylated capture antibodies, according to manufacturer's instructions. The assay was performed according to the manufacturer's protocol with overnight incubation of the diluted samples and standards at 4 °C. The electro-chemiluminescence signal (ECL) was detected by MESO QuickPlex SQ 120 plate reader and analysed with Discovery Workbench Software (v4.0, MSD). The concentration of each sample was calculated based on the parameter logistic fitting model generated with the standards. The concentration was determined according to the certificate of analysis provided by MSD.

## RNA isolation and qRT-PCR

For RNA extraction, we used the PureLink RNA mini Kit (Invitrogen 12183018 A) according to the manufacturer's instructions. mRNA was reverse-transcribed with Superscript Vilo cDNA synthesis kit (Thermo Fisher Scientific, 11754-050). qPCR was performed using SYBR green (Bio-rad, 1725271). The CT values obtained for the genes of interest were corrected for the cDNA input by normalization to the CT value of GAPDH (ΔCT). Furthermore, the ΔCT value was normalized to the mean of control HDs or non-transduced cells. Finally, by using the formula $2^{-\Delta\Delta CT}$, the relative quantification of target cDNA was described as a fold-increase above control and normalized to GAPDH. For each

independent experiment, we averaged the technical triplicate values and this mean was used to calculate the IFN score as the median values of 6 ISG (*IFI27*, *IFI44L*, *IFIT1*, *ISG15*, *RSAD2* and *SIGLEC1*).

To evaluate UPR-dependent genes (*HSPA5*, *ATF4*, and *DDIT3*), THP-1 cells were treated for 6 hrs with 10 μM of thapsigargin or DMSO (vehicle). The RNA extraction, subsequent cDNA synthesis and analyses were performed as previously described. The graphs show -$\log_2$(fold change) relative to non-stimulated HDs fibroblasts or non-infected (NI) THP-1 cells.

The list and sequences of the probes used is supplied in Supplementary Table 2.

## Biochemistry

Protein extraction for whole cell lysate analysis in fibroblasts or THP-1 cells expressing stable CDC42 variants (Supplementary Fig. 3) was performed in RIPA buffer, supplemented with 1% protease inhibitor (Roche, #48679800) and DNAse Benzonase (Merck, #E10145K). 15 μg protein lysate was resolved on Bolt™ 4-12% Bis-Tris Plus Wedgewell gels (Invitrogen, # NW04122BOX). Proteins were subsequently transferred to a polyvinylidene difluoride (PVDF) membrane (Thermo Fisher Scientific, #88518). Membranes were blocked for 1 h with 5% milk solution diluted in TBS−0.1% Tween-20 and further incubated overnight with primary antibodies diluted in the TBS−5% BSA (Sigma, #A7030-500) buffer. The antibodies used were described in Supplementary Table 3. Primary antibodies (1 μg/mL) were revealed with HRP-conjugated secondary antibodies (1 μg/mL) diluted in TBS−0.1% Tween-20, for 1 h at room temperature. Bands were detected by using SuperSignal West Pico Plus chemiluminescence (Thermo Fisher Scientific, #34580). Signals were detected using the Fusion FX imaging system from Vilber.

## Imaging

Images were obtained with an inverted fluorescence microscope (Eclipse Nikon TE300), a Photometrics Cascade camera, and acquired using the Metamorph v7.8.9.0 software. A 100X or 40X objective was used for THP-1 cells and fibroblasts, respectively.

Acquisition of microscopy images were performed as follows: Nuclei staining was used to initially find the focal plane and the cells on coverslips. Then, automatic acquisitions of the corresponding fields were captured for all the required wavelengths. Cells were not selected and all cells present in each microscopy field were subjected to analysis using the Fiji (Fiji is just ImageJ) software (ImageJ version 1.51 u). Overall, at least 15 cells from >10 fields were analysed. The Pearson's Coefficient was measured in Fiji by using a macro containing the Coloco2 plugin as described[11,21]. This coefficient measures the degree of overlap between two stainings. A Pearson's Coefficient value of 0 means that there is no colocalization between the two stainings. By contrast, a value of 1 indicates that there is a perfect colocalization between the two stainings under study. Fibroblasts staining intensity was measured in the whole cells by taking into account the cell morphology identified by F-actin phalloidin staining. P-IRF3 staining intensity in THP-1 cells was measured by focusing and limiting the quantification of the staining to the nucleus.

## Statistics and reproducibility

Statistical analyses were performed using the GraphPad Prism 10.0.2 software. Results are shown as means +/- SEM and the significance levels were calculated using one-way or two-way ANOVA (*$P < 0,05$; **$P < 0,01$; ***$P < 0,001$; ****$P < 0,0001$). Each experiment was repeated independently at least three times.

## Reporting summary

Further information on research design is available in the Nature Portfolio Reporting Summary linked to this article.

## Data availability

The authors confirm that all relevant data are available in the paper and/or its Supplementary Information files or from the corresponding author upon request. Source data are provided with this paper.

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

## Acknowledgements

We thank: the patients and healthy donors for their participation in this study; Didier Busso (CIGEx facility), Sébastien Jacques and Franck Letourneur (GENOM'IC facility), Souganya Many and Yousef Hadjou (CYBIO cell sorting facility) for technical assistance; Gilles Renault (PIV facility) for expertise in statistics; Thomas Henry, Serge Benichou and Marie-Louise Frémond for sharing plasmids and cells; Nadège Bercovici, Clotilde Randriamampita, Fabienne Régnier, Paolo Pierobon and Philippe Mertz for helpful discussions. AIa is supported by an international PhD fellowship from *Université Paris Cité*, a European Society for Immunodeficiencies (ESID) fellowship and by *Fondation pour la Recherche Médicale* (FRM Grant number FDT202404018275). SD is supported by the Research Foundation—Flanders (FWO Grant number 11F4421N). IM is a senior clinical investigator at FWO Vlaanderen (supported by a KU Leuven C1 Grant C16/18/007, by FWO Grant G0B5120N and by the Jeffrey Modell Foundation). JD is supported by Inserm. This project was supported by Inserm, *Centre National de la Recherche Scientifique*, *Université Paris Cité*, *Agence Nationale de la Recherche* (2019, RIDES; 2023 Cytoskinflam), AIRC (IG 28768) and Italian Ministry of Health (5×1000 2019 and PNRR-MR1-2022-12376811).

## Author contributions

J.D. designed and supervised the project. J.D. and I.M. provided funding. A.Ia. and R.T. conducted the experiments. R.E.M. performed the structural characterization. S.D., G.P., M.N.-I., R.T.A.v.W., F.B., A.A.d.J.R., S.C., P.L.A.v.D., A.In., R.G.-M., T.Y., M.T. and I.M. provided samples or cells from CDC42 patients. A.Ia. and J.D. wrote the first draft of the manuscript with contributions from S.D. and I.M. All authors edited the paper.

## Competing interests

The authors declare no competing interests.

## Additional information

[1]Université Paris Cité, CNRS, Inserm, Institut Cochin, F-75014 Paris, France. [2]Laboratory for Inborn Errors of Immunity, KU Leuven, Leuven, Belgium. [3]Department of Pediatrics, University Hospitals Leuven, Leuven, Belgium. [4]Laboratory of Immuno-Rheumatology, Bambino Gesù Children's Hospital, IRCCS, Rome, Italy. [5]Department of Pediatrics, Kyoto University Graduate School of Medicine, Kyoto, Japan. [6]Department of Pathology & Clinical Bioinformatics, Erasmus University Medical Center, Rotterdam, The Netherlands. [7]Translational Autoinflammatory Disease Section (TADS), National Institute of Allergy and Infectious Diseases (NIAID), National Institutes of Health, Bethesda, MD 20892, USA. [8]National Center for Rare Diseases, Istituto Superiore di Sanità, 00161 Rome, Italy. [9]Department of Internal Medicine, Division of Allergy & Clinical Immunology, Erasmus University Medical Center, Rotterdam, The Netherlands. [10]Department of Immunology, Erasmus University Medical Center, Rotterdam, The Netherlands. [11]Division of Rheumatology, ERN RITA Center, IRCCS Ospedale Pediatrico Bambino Gesù, Rome, Italy. [12]Molecular Genetics and Functional Genomics, Bambino Gesù Children's Hospital, IRCCS, 00146 Rome, Italy. [13]Present address: Department of Cell Physiology & Metabolism, Faculty of Medicine, University of Geneva, Geneva, Switzerland. ✉e-mail: jerome.delon@inserm.fr

