## [Transparent Peer Review file · Nature Communications]

Autoinflammatory patients with Golgi-trapped CDC42 exhibit intracellular trafficking defects leading to STING hyperactivation and ER stress

Corresponding Author: Dr Jerome Delon

Version 0:

Reviewer comments:

Reviewer #1

(Remarks to the Author)

The manuscript by Iannuzzo et al. investigates the mechanisms underlying autoinflammation in patients that express mutant CDC42 proteins. Interestingly, the authors established that two variants, of the four discovered and linked to autoinflammation, are trapped in the Golgi blocking anterograde and retrograde transport, favoring Golgi accumulation of STING and resulting in STING hyperactivation. Importantly, the authors show that the expression of these mutant proteins increases expression of six IFN-stimulated genes and this is mediated by STING. Thus, the authors conclude that STING hyperactivation could be responsible for autoinflammation thus opening the way to the identification of novel therapeutical means.

The manuscript is well-written, experiments are well-executed with appropriate controls and results are interesting. Indeed, the authors investigated and discovered the mechanism by which two CDC42 mutant proteins are responsible for autoinflammation. The conclusion of the authors are fully supported by the data shown.

Minor points

-The manuscript describes the use of cell lines (THP1, HEK-293T) but does not include relevant information such as, for instance, accession numbers provided by a database where the genetic information is deposited. Moreover, it is not clear if these cells have been authenticated to ensure their identity.

-Typos should be corrected (for instance "pennicillin" should be "penicillin").

-In the discussion the authors should talk about the possible strategy to reduce STING hyperactivation in the patients.

Reviewer #2

(Remarks to the Author)

The work by Iannuzzo explores the role of CDC42 mutants in STING activation due to their trafficking defects.

The major discoveries that served as a foundation for this work were made by Nishitani-Isa in their excellent paper published in the JEM in 2022. Therefore, I feel that even though the current work extends the earlier study, it is not sufficiently novel.

I also found the microscopy data unconvincing. There is no quantitative analysis of wild-type and mutant CDC42 overexpression in parental and THP-1 cells (by the Western), and it is unclear whether mutant protein function can be attributed to their overexpression.

While the Pearson's correlation coefficient difference between 0.5 and 0.7 is statistically significant, is this difference large enough to ascribe mutant protein function to mislocalization?

The number of samples in different experiments is substantially different, varying from 3 to 14 (in Figure 5A) – why? Is there overpowering of statistics? Plotting all the values of individual biological replicates would be useful.

The standardization of controls (in several instances) to 100% and then using these values for the analysis is not

appropriate, as this suggests the lack of variation across control samples, eg Figure 4, 5E, and 5G. Figure 5G is particularly surprising, as a significant difference is claimed between 100 and <110 (OY axis starts at 80). This difference is within the operator's error and is almost impossible to be biologically relevant.

Instead of quantifying several cells to establish relative levels of expression of one or another protein, it would be better to assess this using flow cytometry or the Western, where thousands of cells can be processed.

"Randomly-imaged cells" are not appropriate – the microscopy should be double-blinded.

All in all, this is a potentially interesting paper, but I find the robustness and the relative lack of novelty of these studies quite problematic.

Reviewer #3

(Remarks to the Author)

In this manuscript, Iannuzzo and colleagues provide mechanistic insight into the markedly different phenotypes seen in patients with pathogenic variants in CDC42. Specifically, why patients with specific C-terminal variants including R186C and *192C*24 present with the autoinflammatory NOCARAH syndrome, while patients with the p.Y64C variant, located closer to the N-terminus, present with a different constellation of symptoms including intellectual delay, growth retardation and dysmorphic facial features, macrothrombocytopenia.

Prior reports have demonstrated that c-terminal variants vary in degree of accelerated CDC42 degradation, prenylation, palmitoylation, and cellular localization and these variants may induce signaling within a number of inflammatory pathways with excessive production of IL-1, IL-6, TNF, IL-18, IFN-g, and CXCL9.

Here, the authors delve deeper into implications of the distinct subcellular localization of CDC42 mutants, with two of the c-terminus variants (R186C and *192C*24) localized to the Golgi. They demonstrate that the aberrant localization is due to a block in bidirectional ER-Golgi trafficking. The Golgi-trapped CDC42 induces ER stress as well as hyperactivation of the STING pathway leading to an elevated type I interferon signature.

The paper and figures are very clearly written and provide novel mechanistic insight into NOCARAH syndrome and into CDC42 function. I would like to see the paper published after the following modifications.

-ER Stress should be more extensively characterized. The BIP immunofluorescence outlined in figure 4 should be complemented by additional markers of ER stress in patient fibroblasts and THP-1 lines expressing pathogenic CDC42 variants. qPCR evaluation of UPR-dependent genes such as HSPA5, ATF4, and DDIT3 before and after thapsigargin would provide this characterization. Note that the authors of this paper used this approach to characterize increased ER stress induced by heterozygous variants in the C-terminal domain of COPA. (PMID: 38175705)

Version 1:

Reviewer comments:

Reviewer #1

(Remarks to the Author)

The authors answered satisfactorily to my criticisms. Thus I recommend publication of the manuscript.

Reviewer #2

(Remarks to the Author)

Iannuzzo et al have responded to many of my queries.

1. I accept their explanation.

2. I appreciate that the authors provided additional data and clarification of the methods. They are correct that some of the information was included in the figure legends, and I apologise for overlooking this. However, I believe that providing clarification in the Methods section is valuable.

- I did not fully understand the use of four healthy donor cells. Did the authors mean four donors? Their reference to other (flawed) papers showing a single control cell is irrelevant and unhelpful.

- Please clarify the matched expression of constructs. If I understood correctly, they sorted overexpressing cells based on their identical GFP fluorescence intensity; is this correct? Also, please clarify the Western blot results – different mutants show different expression levels and profiles. Is this due to their instability, variable glycosylation, phosphorylation, or mislocalisation?

- The lack of an effect from the overexpression of WT CDC42, as explained in the last two paragraphs of this section, is reasonable. However, the reference to papers by others is unnecessary and does not alter my comments – many experimental practices are incorrect, regardless of where they are published.

3. This is reasonable.

4. This is acceptable.

5. I accept the authors' explanation and appreciate their new remarks in the text. I commend the authors for showing non-

standardised values while still demonstrating statistical significance. For reference, I can provide examples of highly ranked papers where controls were all standardised to the same value, but I appreciate the authors' effort here.
6-8. Fine.

Reviewer #3

(Remarks to the Author)

The authors have adequately addressed my concerns with their additional data and modifications to the manuscript.

Point-by-point responses to the Reviewers for the manuscript NCOMMS-24-05109 by Iannuzzo *et al.* entitled “Autoinflammatory patients with Golgi-trapped CDC42 exhibit intracellular trafficking defects leading to STING hyperactivation and ER stress”

Reviewer #1 (Remarks to the Author):

The manuscript by Iannuzzo *et al.* investigates the mechanisms underlying autoinflammation in patients that express mutant CDC42 proteins. Interestingly, the authors established that two variants, of the four discovered and linked to autoinflammation, are trapped in the Golgi blocking anterograde and retrograde transport, favoring Golgi accumulation of STING and resulting in STING hyperactivation. Importantly, the authors show that the expression of these mutant proteins increases expression of six IFN-stimulated genes and this is mediated by STING. Thus, the authors conclude that STING hyperactivation could be responsible for autoinflammation thus opening the way to the identification of novel therapeutical means. The manuscript is well-written, experiments are well-executed with appropriate controls and results are interesting. Indeed, the authors investigated and discovered the mechanism by which two CDC42 mutant proteins are responsible for autoinflammation. The conclusion of the authors are fully supported by the data shown.

Minor points

-The manuscript describes the use of cell lines (THP1, HEK-293T) but does not include relevant information such as, for instance, accession numbers provided by a database where the genetic information is deposited. Moreover, it is not clear if these cells have been authenticated to ensure their identity.

We thank the Reviewer for this remark.

ATCC accession numbers are TIB-202 and CRL-11268 for THP-1 (Page 16 line 356) and HEK-293T (Page 16 line 353) cell lines, respectively. We have added this information in the text.

These cells were authenticated originally by the supplier so we have not authenticated them again. The same THP-1 cells were transfected or transduced in parallel with WT or mutant CDC42 constructs, and functional consequences were studied at the same time. Thus, each construct was expressed in the context of the same genetic background (isogenic lines). The HEK-293T cells were also the same line used to produce in parallel WT or mutant CDC42 viruses to transduce the same day the THP-1 cell line used in the present study.

-Typos should be corrected (for instance "pennicillin" should be "penicillin").

We apologise for this mistake. We have now corrected it in the text (Page 16 lines 355 and 360). We have also run additional spelling checks.

-In the discussion the authors should talk about the possible strategy to reduce STING hyperactivation in the patients.

We thank the Reviewer for this suggestion. As the results we report here open indeed the way to the identification of novel therapeutic means, we agree that this aspect was lacking in the *Discussion*.

As far as we know, the H-151 STING inhibitor, which has largely been used *in vitro*, is not used in clinics. Novartis has developed an inhibitor but there is no published data on humans so far. The mainstay of treatment for decreasing the activation of the STING - Type I IFN pathway has been to use JAK inhibitors such as Ruxolitinib or Baricitinib, but they block many cytokine receptors so they are not very specific (PMID: 27554814; PMID: 29649002; PMID: 34375617; PMID: 37741518). Due to this critical issue, inhibitors with increased specificity have recently been developed, including Anifrolumab, an anti- type I IFN receptor (IFNAR) monoclonal antibody, which seems to provide better efficacy (PMID: 38373653, PMID: 37640261). As an alternative approach, targeted STING degradation (PMID: 38811577) appears to be valuable for treating patients presenting a hyperactivation of the STING - Type I IFN pathway.

We have now added these different points in the Discussion (Pages 13-14 lines 303-313).

Reviewer #2 (Remarks to the Author):

The work by Iannuzzo explores the role of CDC42 mutants in STING activation due to their trafficking defects.

The major discoveries that served as a foundation for this work were made by Nishitani-Isa in their excellent paper published in the JEM in 2022. Therefore, I feel that even though the current work extends the earlier study, it is not sufficiently novel.

We politely disagree with the Reviewer's remark. The cited paper (PMID: 35482294) is indeed an excellent paper that demonstrates that Golgi-trapped CDC42 variants overactivate the Pyrin inflammasome. Drs Masahiko Nishitani-Isa and Takahiro Yasumi, leading authors of that paper, are co-authors of the present manuscript because they provided biological samples required to assess the IFN α levels and determine the occurrence of a type I interferon signature. This collaboration also allowed us to mechanistically explore the involved players/circuits. In their original JEM paper, they claimed that COP-I-mediated vesicular transport and STING subcellular localisation were not impaired by Golgi-trapped CDC42 variants in COS-1 cells overexpressing CDC42 variants. By contrast, in the current paper, using patients' cells and careful quantifications of microscopy images, we clearly show here that the recurrent R186C variant specifically induces a block in bidirectional ER-Golgi trafficking (Fig. 2 and 3) and STING enrichment in the Golgi (Fig. 5A). While it is unclear, at the moment, to what extent the use of different cell types could be responsible for this discrepancy, the present report is therefore the first establishing such defects in intracellular protein transports and STING localisation elicited by CDC42 R186C, providing relevant mechanistic insights. Using the innate immune THP-1 cells, we also show a unique causal effect of the Golgi-trapped CDC42 variants on STING accumulation in the Golgi in a COPI-dependent manner. Our data, based on the use of complementary cellular systems, are thus fully consistent. We also provide evidence of the hyperactivation of the STING pathway as shown by increased P-IRF3, P-STAT1 and IFN score observed specifically in patients' fibroblasts and THP-1 cells expressing Golgi-trapped CDC42 variants. This is also confirmed by the high levels of IFN α we measured in blood samples from 5 out of 6 unrelated R186C and *192C*24 CDC42 patients (Fig. 7). Furthermore, we have now provided additional data that reinforce a role for Golgi-trapped CDC42 variants in inducing strong ER stress, by analysing both BiP expression levels (Fig. 4A-C) and the amounts of 3 mRNA involved in the unfolded protein response (UPR) (Fig. 4D, E) in patients' and THP-1 cells.

Altogether, by establishing that Golgi-trapped CDC42 variants specifically block bidirectional ER-Golgi protein transport which in turn induce ER stress and hyperactivate the STING - Type I IFN pathway, we provide novel insights that had not been considered by the Nishitani-Isa paper focused on the Pyrin inflammasome. We hope this convinces the Reviewer of the extent of novelty.

I also found the microscopy data unconvincing. There is no quantitative analysis of wild-type and mutant CDC42 overexpression in parental and THP-1 cells (by the Western), and it is unclear whether mutant protein function can be attributed to their overexpression.

We are sorry for the tepid feedback and regret that the Reviewer was unconvinced. We have a recognized expertise in imaging and quantification of microscopy experiments. We

accurately quantified each microscopy data as reported before (see our previous articles in JACI 2020 and JCI 2024). This is compulsory because, with this kind of single cell approach, one quickly realises that each cell is actually unique, and there is a certain dispersion of behaviour. Here, instead of showing one “representative” experiment, we have plotted the mean value of the quantifications from all the microscopy experiments we have performed. Thus, one dot is the mean value of about 15 individual cells from one independent experiment. To be clearer, we have added more detailed information about image analysis procedures in the Methods section of the revised version (Page 21 lines 482-484).

Furthermore, many articles in this field use one single control cell from one healthy donor. We have used 4 different unrelated healthy donor cells for the microscopy data in order to account for the inherent variability. In the end, it turned out that the intra-control variability was quite low here.

In addition, main results obtained by microscopy such as specific increases in BiP expression and STAT1 phosphorylation in cells expressing Golgi-trapped CDC42 variants have now been shown to be also true using flow cytometry acquisition of 10,000 cells and Western blot (see below and Fig. 4B, 4C and 6D).

Moreover, we believe that microscopy is the best technique to study intracellular protein trafficking, and we have validated this method in our JCI paper published earlier this year (Delafontaine *et al.*, PMID: 38175705).

In any case, all raw microscopy data are fully available.

We had also carefully quantified the expression levels of the different WT and variants of CDC42 in order to achieve matched expression of these different forms of CDC42 in THP-1 cells. This is compulsory in order to ascribe the function of one particular variant. As previously explained in the Methods section, we used a bicistronic lentiviral vector that encodes for both GFP and Myc-tagged WT CDC42 or variants (Page 13 lines 296-298 in the original manuscript). The GFP fluorescence was used to sort THP-1 cells expressing matched levels of GFP (see the FACS strategy in Supplementary Fig. 1B). We also checked that expression of Myc-tagged CDC42 was similar between WT and variants. We now provide these flow cytometry profiles in the Supplementary Fig. 2B:

Western blot experiments also show similar expression of Myc-tagged CDC42 for all the variant forms (Supplementary Fig. 3):

Importantly, the very same strategy was used in the Nishitani-Isa paper mentioned above. Using THP-1 cells overexpressing WT or different variants of CDC42, they show that only the two Golgi-localised CDC42 variants can induce an excess of pyroptosis compared to WT or other CDC42 variants (Fig. 7A-D). From these experiments, they rightly attributed a specific role of these two CDC42 variants in Pyrin hyperactivation, which is actually the main message of their paper.

Here, also in THP-1 cells, we have applied the same approach and have systematically compared the functions of each CDC42 variant to the function of WT CDC42. This is a very classic approach performed not only in the JEM paper by Nishitani-Isa *et al.* but also in many other excellent articles that have overexpressed other gene variants in THP-1 cells to attribute specific functions to these variants (see for example the recent papers by:

- David *et al. J Exp Med*, PMID: 38869500 in Figures 4B-F and S3C-F, J, K
 - Hirschenberger *et al. Nat Commun*, PMID: 37914730 in Figure 3a
 - Al-Azab *et al. Nat Immunol*, PMID: 38831104 in Figures 3a-e, 4e, 5a-e and Extended Data Figures 4a-f, 5ab, 6d, 7a-e
- , all in THP-1 cells).

In addition, all published prominent papers about CDC42 patients have all used the classic approach of overexpressing WT or mutated CDC42 in multiple cell lines (HEK 293T, COS-1, COS-7 and NIH-3T3) to test specific defects of CDC42 variants in CDC42 activity, binding to Rho-GDI and effectors, palmitoylation, subcellular localization, protein expression/stability, post-translational modifications, and migration (Lam *et al. J Exp Med*, PMID: 31601675; Nishitani-Isa *et al. J Exp Med*, PMID: 35482294; Coppola *et al. J Allergy Clin Immunol*, PMID: 35157921). From all these overexpression experiments, the field has gained crucial knowledge in the understanding of the mechanisms responsible for the pathogenicity of CDC42 variants.

Furthermore, depending on the experiments, we also had data for non-transfected, non-infected or GFP-transfected cells, so that we have information regarding a possible functional effect coming from WT CDC42 overexpression. We did not observe any effect of WT CDC42 overexpression on BiP expression (Fig. 4C), UPR-dependent genes (Fig. 4E), STING accumulation in the Golgi (Fig. 5B, C), levels of P-IRF3 (Fig. 6A) and P-STAT1 (Fig. 6D), and on the IFN scores (Fig. 8B-E).

Finally, the data on THP-1 cells are fully consistent with what we observed in patients' cells carrying heterozygous CDC42 variants, thus also expressing a pool of WT CDC42 in the absence of overexpression.

While the Pearson's correlation coefficient difference between 0.5 and 0.7 is statistically significant, is this difference large enough to ascribe mutant protein function to mislocalization? The microscopy experiments to study the bidirectional intracellular protein transport (Fig. 1 and 2) show Pearson's coefficients to be around 0.5 or 0.7 for two groups of cells: the 4 controls and Y64C on one hand, and the R186C on the other hand. Clearly, in the time frame of these experiments, the R186C patient's cells show a block in both anterograde and retrograde transports because the Pearson's coefficients remain unchanged only for this variant between the initial and the final time points of these experiments. These differences in Pearson's Coefficients values appear to have functional consequences related to a block in intracellular transport as described for COPA patients. We show indeed that only the R186C patient's cells exhibit stable PC-1 expression (Fig. 2D), higher ER stress (Fig. 4A, B) and UPR-dependent genes (Fig. 4D), stronger accumulation of STING in the Golgi (Fig. 5A), higher phosphorylations of IRF3 and STAT1 (Fig. 6B, C) and IFN score (Fig. 8A).

The number of samples in different experiments is substantially different, varying from 3 to 14 (in Figure 5A) – why? Is there overpowering of statistics? Plotting all the values of individual biological replicates would be useful.

We initially had access only to R186C patient's and control cells. Thus, the first experiments were performed only with this variant. When we later obtained cells from the Y64C patient, we added this condition in parallel with the CTL and R186C conditions. The same applies for the experiments with the THP-1 cells because we obtained constructs for Y64C, C188Y and *192C*24 variants after the R186C ones. Altogether, this explains why there are sometimes more data points with the R186C variant compared to the other ones. Instead of discarding the initial experiments, we felt it was more honest to show all the experiments performed.

We apologise for Fig. 5A which is indeed unclear. Different HD cells had been pooled together in the original figure. We have now corrected that:

It shows that R186C patient's cells exhibit a higher localisation of STING in the Golgi compared to each 4 HDs and Y64C cells.

The standardization of controls (in several instances) to 100% and then using these values for the analysis is not appropriate, as this suggests the lack of variation across control samples, e.g. Figure 4, 5E, and 5G. Figure 5G is particularly surprising, as a significant difference is

claimed between 100 and <110 (OY axis starts at 80). This difference is within the operator's error and is almost impossible to be biologically relevant.

The Reviewer raised a fair remark. However, standardisation to 100 or 1 might be required when control values are very different from one independent experiment to another, for example because flow cytometry or microscopy acquisition parameters are different between individual experiments (See recent examples: Fig. 1a and 1c in Durand *et al. Nat Commun*, PMID: 38409097; Fig. S3A in David *et al. J Exp Med*, PMID: 38869500; Fig. 4A, 4C, 5B, 6I and 7D in Simula *et al. Nat Commun*, PMID: 38467616).

The data presented in the original Fig. 5G falls in this situation. The first two experiments were performed to compare P-STAT1 levels in control and R186C fibroblasts. Months later, we obtained fibroblasts from the Y64C patient, so we conducted four additional experiments by comparing side by side the steady state P-STAT1 levels between CTL, Y64C and R186C cells. Unfortunately, between these two periods, the acquisition settings were different. Nevertheless, R186C-expressing cells systematically exhibited higher levels of P-STAT1 in all six independent experiments compared to HD cells, making it unlikely that such results could be due to the operator's error:

The new figure without any standardisation is now shown in Fig. 6C:

However, it is absolutely true that the difference in P-STAT1 levels between CTL and R186C cells is very small in these steady state conditions. We have indicated that in the text (Page 10 lines 221-222). Using flow cytometry, we have now revisited and extended this part in the new version of the manuscript. We show that Golgi-trapped CDC42 variants induce an excess

of STAT1 phosphorylation upon IFN α stimulation of THP-1 cells (Page 10 lines 222-225) (Fig. 6D):

In addition, the absence of standardisation in the original Fig. 4 (now Fig. 4A) still shows that R186C patient's cells exhibit higher BiP expression than HD and Y64C cells:

The same is true for Fig. 5E (now Fig. 6A). The absence of standardisation still shows that R186C and *192C*24 cells have higher P-IRF3 levels than WT cells:

Altogether, the absence of standardisation now shown in these figures still indicates that R186C and *192C*24 cells, but not Y64C or C188Y ones, exhibit statistically significant and unique functional features compared to control cells.

Instead of quantifying several cells to establish relative levels of expression of one or another protein, it would be better to assess this using flow cytometry or the Western, where thousands of cells can be processed.

We thank the reviewer for the suggestion.

We have now added quantifications of BiP expression using:

- Western blot (Fig. 4B):

We confirm here that R186C fibroblasts express high basal level of BiP, as shown initially by microscopy (Fig. 4A). Interestingly, BiP expression does not increase much upon thapsigargin treatment, suggesting that basal BiP expression is maximal in R186C fibroblasts.

- Flow cytometry (acquisition of 10,000 cells) (Fig. 4C):

Only R186C and *192C*24 CDC42 variants exhibit BiP overexpression upon thapsigargin treatment.

We have also used flow cytometry to quantify STAT1 phosphorylation (see above, Fig. 6D).

Thus, these new figures panels, obtained with a different technique, are fully in agreement with the results originally obtained by microscopy.

“Randomly-imaged cells” are not appropriate – the microscopy should be double-blinded. We thank the Reviewer for the suggestion to clarify the term “randomly-imaged cells”. Acquisitions of microscopy images were performed as follows: Nuclei stainings were used to initially find the cells and the focal plane on coverslips. Then, automatic acquisitions of the corresponding fields were captured for all the required wavelengths. Cells were not selected and all cells present in a microscopy field were subjected to quantitative analysis. This is what we meant by “randomly-imaged cells”. We have now further explained our acquisition procedure in the Methods section (Page 21 lines 478-482). Given the large amount of imaging data we generated, it was impossible to have two full time dedicated people to work systematically side by side to achieve double-blinding. Thus, we have removed the mention of “randomly-imaged cells” in the revised manuscript to avoid any confusion.

All in all, this is a potentially interesting paper, but I find the robustness and the relative lack of novelty of these studies quite problematic.

We have addressed above all the constructive remarks raised by the Reviewer, so we hope that our answers and the additional experiments performed during the allowed 3-month revision period have increased the robustness of the data. Of note, our use of THP-1 cells overexpressing CDC42 variants has allowed us to attribute specific functions to variants that share similar localisation defects. This is the very same strategy used in the Nishitani-Isa paper mentioned above, and in many other studies. Furthermore, all the new results obtained with new techniques have confirmed all the findings described in the original manuscript. Regarding the novelty, we show here that Golgi-trapped CDC42 variants induce unique functional defects. They impair the bidirectional ER-Golgi intracellular protein transport and favour STING accumulation in the Golgi in a COPI-dependent manner. Consequently, the STING pathway is hyperactivated as shown by increases in the phosphorylation of both IRF3 and STAT1, ISG expression and IFN α levels. Mirroring these events, we also show that Golgi-trapped CDC42 variants induce a strong ER stress as demonstrated by high expression of the BiP protein and genes of the Unfolded Protein Response. These are new results that have not been reported previously.

Reviewer #3 (Remarks to the Author):

In this manuscript, Iannuzzo and colleagues provide mechanistic insight into the markedly different phenotypes seen in patients with pathogenic variants in CDC42. Specifically, why patients with specific C-terminal variants including R186C and *192C*24 present with the autoinflammatory NOCARAH syndrome, while patients with the p.Y64C variant, located closer to the N-terminus, present with a different constellation of symptoms including intellectual delay, growth retardation and dysmorphic facial features, macrothrombocytopenia.

Prior reports have demonstrated that c-terminal variants vary in degree of accelerated CDC42 degradation, prenylation, palmitoylation, and cellular localization and these variants may induce signaling within a number of inflammatory pathways with excessive production of IL-1, IL-6, TNF, IL-18, IFN- γ , and CXCL9.

Here, the authors delve deeper into implications of the distinct subcellular localization of CDC42 mutants, with two of the c-terminus variants (R186C and *192C*24) localized to the Golgi. They demonstrate that the aberrant localization is due to a block in bidirectional ER-Golgi trafficking. The Golgi-trapped CDC42 induces ER stress as well as hyperactivation of the STING pathway leading to an elevated type I interferon signature.

The paper and figures are very clearly written and provide novel mechanistic insight into NOCARAH syndrome and into CDC42 function. I would like to see the paper published after the following modifications.

-ER Stress should be more extensively characterized. The BiP immunofluorescence outlined in figure 4 should be complemented by additional markers of ER stress in patient fibroblasts and THP-1 lines expressing pathogenic CDC42 variants. qPCR evaluation of UPR-dependent genes such as HSPA5, ATF4, and DDIT3 before and after thapsigargin would provide this characterization. Note that the authors of this paper used this approach to characterize increased ER stress induced by heterozygous variants in the C-terminal domain of COPA. (PMID: 38175705)

We thank the Reviewer for the suggestion. We have now largely expanded our ER stress studies in a new version of Figure 4.

In addition to the initial data shown on patients' fibroblasts using microscopy, we have now also quantified BiP expression by Western blot to address one comment from Reviewer 2. In a new Figure 4B, we confirm that R186C fibroblasts express high basal level expression of BiP. Interestingly, BiP expression does not increase much upon thapsigargin treatment, suggesting that basal BiP expression is maximal in R186C fibroblasts:

We have also measured BiP levels upon thapsigargin treatment by flow cytometry in THP-1 cells expressing CDC42 variants. The results clearly show that only the two Golgi-trapped R186C and *192C*24 CDC42 variants induce higher BiP expression (Fig. 4C):

Furthermore, we have also performed qPCR experiments to evaluate the expression of three UPR-dependent genes (*HSPA5*, *ATF4*, and *DDIT3*), as performed recently in our *COPA* article.

In our hands, fibroblasts do not express DDIT3 mRNA but very high levels of both HSPA5 and ATF4 mRNA were measured only in CDC42 R186C cells upon thapsigargin treatment (Fig. 4D):

Similarly, all three UPR-dependent genes were highly expressed only in THP-1 cells expressing the two Golgi-trapped R186C and *192C*24 CDC42 variants after thapsigargin treatment (Fig. 4E). Of note, basal levels of DDIT3 mRNA levels were also very high only in both R186C and *192C*24 CDC42 variants:

Altogether, all these new results reinforce the initial findings of a specific effect of Golgi-trapped CDC42 variants in inducing a strong ER stress.

Point-by-point responses to the Reviewers for the manuscript NCOMMS-24-05109 by Iannuzzo *et al.* entitled “Autoinflammatory patients with Golgi-trapped CDC42 exhibit intracellular trafficking defects leading to STING hyperactivation and ER stress”

Reviewer #1 (Remarks to the Author):

The authors answered satisfactorily to my criticisms. Thus I recommend publication of the manuscript.

We thank the Reviewer for the suggestions that significantly improved the revised version of the manuscript.

Reviewer #2 (Remarks to the Author):

Iannuzzo et al have responded to many of my queries.

1. I accept their explanation.

2. I appreciate that the authors provided additional data and clarification of the methods. They are correct that some of the information was included in the figure legends, and I apologise for overlooking this. However, I believe that providing clarification in the Methods section is valuable.

We thank the Reviewer for the comments.

- I did not fully understand the use of four healthy donor cells. Did the authors mean four donors? Their reference to other (flawed) papers showing a single control cell is irrelevant and unhelpful.

We used indeed four different control cells from four unrelated healthy donors to account for the intrinsic variability in control patients. In the end, the intra-control variability was found to be quite low in each experiment.

- Please clarify the matched expression of constructs. If I understood correctly, they sorted overexpressing cells based on their identical GFP fluorescence intensity; is this correct?

Yes, the lentiviral vector consists of the Myc-CDC42 and GFP sequences separated by an IRES cassette. Cells were sorted based on GFP expression and also stained for anti-Myc and analyzed by FACS for quantifying the expression of Myc-CDC42 WT or mutant forms.

Also, please clarify the Western blot results – different mutants show different expression levels and profiles. Is this due to their instability, variable glycosylation, phosphorylation, or mislocalisation?

It is true that the different CDC42 variants show, by Western blot, slightly varying levels of expression. We believe that the most plausible explanation lies indeed on mislocalization of many of these variants that are not present in the same subcellular compartments, and thus are more likely to exhibit different levels of detergent solubility. Of note, the Myc intracellular staining and FACS analysis provided more homogenous expression levels of the different forms of CDC42.

- The lack of an effect from the overexpression of WT CDC42, as explained in the last two paragraphs of this section, is reasonable. However, the reference to papers by others is unnecessary and does not alter my comments – many experimental practices are incorrect, regardless of where they are published.

3. This is reasonable.

4. This is acceptable.

5. I accept the authors' explanation and appreciate their new remarks in the text. I commend the authors for showing non-standardised values while still demonstrating statistical significance. For reference, I can provide examples of highly ranked papers where controls were all standardised to the same value, but I appreciate the authors' effort here.

6-8. Fine.

We thank the Reviewer for the suggestions that significantly improved the revised version of the manuscript.

Reviewer #3 (Remarks to the Author):

The authors have adequately addressed my concerns with their additional data and modifications to the manuscript.

We thank the Reviewer for the suggestions that significantly improved the revised version of the manuscript, especially for the ER stress experiments.